# Study of Structural, Compression, and Soft Magnetic Properties of Fe_65_Ni_28_Mn_7_ Alloy Prepared by Arc Melting, Mechanical Alloying, and Spark Plasma Sintering

**DOI:** 10.3390/ma16227244

**Published:** 2023-11-20

**Authors:** Kaouther Zaara, Virgil Optasanu, Sophie Le Gallet, Lluisa Escoda, Joan Saurina, Frédéric Bernard, Mohamed Khitouni, Joan-Josep Suñol, Mahmoud Chemingui

**Affiliations:** 1Department of Physics, University of Girona, Campus Montilivi, 17071 Girona, Spain; kaouther_zaara@hotmail.fr (K.Z.); lluisa.escoda@udg.edu (L.E.); joan.saurina@udg.edu (J.S.); 2Laboratoire Interdisciplinaire Carnot de Bourgogne—ICB UMR 6303 CNRS, Université de Bourgogne, BP 47870, CEDEX, 21078 Dijon, France; virgil.optasanu@u-bourgogne.fr (V.O.); sophie.le-gallet@u-bourgogne.fr (S.L.G.); frederic.bernard@u-bourgogne.fr (F.B.); 3Department of Chemistry, College of Science, Qassim University, Buraidah 51452, Saudi Arabia; kh.mohamed@qu.edu.sa; 4Laboratory of Inorganic Chemistry, LR 17-ES-07, University of Sfax, B.P. 1171, Sfax 3018, Tunisia; chmingui_mahmoud@yahoo.fr

**Keywords:** nanostructure, arc melting, spark plasma sintering, X-ray diffraction, magnetic behavior, mechanical properties, Fe-Ni-Mn alloys

## Abstract

Soft magnetic Fe_65_Ni_28_Mn_7_ (at. %) alloy was successfully synthesized by mechanical alloying and spark plasma sintering (SPS) and, in parallel, the same composition was prepared by arc melting (AM) for comparison. Several SPS conditions were tested. X-ray diffraction and scanning electron microscopy were used to investigate the structure, phase composition, and morphology of the samples. It was found that mechanical alloying produced BCC and FCC supersaturated solid solution after 130 h of milling, with a fine microstructure (i.e., crystallite size of 10 nm). Spark plasma sintering performed at 750 °C and 1000 °C under two pressures of 50 MPa and 75 MPa revealed stable FCC phases. A single FCC phase was observed after the arc melting synthesis. The magnetic properties of milled powders and solids obtained by AM and SPS were investigated. The specimen consolidated by SPS at 1000 °C under the pressure of 50 MPa exhibits soft magnetic behavior (coercivity 0.07 Oe), whereas the mechanically alloyed sample revealed hard magnetic behavior. The specimen consolidated at 750 °C under a pressure of 75 MPa showed a higher compressive strength of 1700 MPa and a Vickers hardness of 425 ± 18 HV. As a result, sintering at 750 °C/75 MPa can be utilized to enhance the mechanical properties, while those sintered at 1000 °C/50 MPa increase magnetic softness.

## 1. Introduction

Mechanical alloying (MA) is a popular and effective process for producing nanocrystalline powders with distinct chemical, physical, and mechanical properties that distinguish them from conventional materials [1,2,3,4,5,6,7,8,9,10]. This process was commonly utilized in the manufacture of soft magnetic alloys as supersaturated solid solutions, multiphase, or possibly amorphous structures [4]. In general, some magnetic and mechanical properties can be improved by reducing crystallite size to the nanoscale, whereas the presence of stresses and defects introduced by mechanical alloying hardens the magnetic behavior; the overall magnetic property is a competition between reducing crystallite size and increasing lattice strain.

Mechanical alloying produces powdered samples. Thus, to produce parts, it is necessary to proceed to powder consolidation and/or sintering. Particularly with nanocrystalline soft magnetic materials, bulk product fabrication is complex. However, a large fabrication of bulk nanocrystalline material is required for commercial applications. The spark plasma sintering (SPS) process is currently increasing in popularity as a novel powder consolidation technique [11,12,13,14,15,16]. By retaining the nanocrystalline or amorphous structure, the SPS approach may restrict grain growth during compaction and enable bulk material production. A suitable ferromagnetic substance is pure iron. It has an extremely low resistivity, which results in significant Eddy current losses.

Regarding magnetic applications, devices utilizing alloyed iron have a higher efficiency than those using pure iron cores because of higher magnetic permeability and lower overall core losses. Furthermore, different alloying components can affect anisotropy and magnetization in different ways [17]. Hardening induced by increased microstrain favors the development of a semi-hard behavior. These materials are suitable for magnetically coupled devices, such as brakes, clutches, and tensioners. The solid solution’s magnetic properties are supposed to change linearly with the weight percentage of elements added in the alloy if two components are assumed to be ferromagnetic and have known saturation magnetization values. Nickel metal is mixed with iron to increase electrical resistivity, which decreases Eddy current loss. Additionally, higher permeability and saturation magnetization are produced by an increase in Ni content [18]. The Fe-Ni alloy combination therefore has good soft magnetic characteristics.

Regarding mechanical response, based on its influence on the strength of Fe grain boundaries, Seah [19] categorized the addition of several elements. According to the author, although Mo and C helped to strengthen the grain boundaries, components like P and Mn caused the borders to weaken. Squires and Wilson [20] hypothesized that embrittlement is caused by Mn atom segregation at the grain boundaries during aging. They deduce it in light of the brittle fracture along the primary austenite grain boundaries and the absence of second particles in the fracture surfaces. Heo [21] has also shown that the irregular segregation of Ni and Mn atoms through grain boundaries in Fe-7Ni-8Mn wt.% alloy results in embrittlement that is consistent with the concept of heat embrittlement [19]. In addition, Steven and Balajiva [22] first described Mn as an element that causes brittleness in alloys. This idea was subsequently supported by Schultz and McMahon [23] and Weng and McMahon [24]. Wilson [25] disagreed, however, that manganese segregates at grain boundaries as a first step in the production of grain boundary precipitates and functions as a significant embrittling component in the early phases of aging.

Therefore, the development of Fe-Ni-Mn bulk alloys to prevent brittleness is a field of scientific and technological interest. Thus, numerous attempts have been made on Fe–Ni–Mn alloys to address their brittleness problems, which can be arranged by the combination of mechanical alloying and SPS techniques. As known, Fe-Ni-Mn alloys are a class of high-strength coating martensitic alloys. They are sufficiently ductile in the solution-annealed state but suffer from severe intergranular embrittlement along preceding austenite grain boundaries [26,27]. Also, martensitic transformation in Fe-Ni-Mn alloys has been widely investigated [28]. The mechanical alloying method was used to produce the Fe_86_Ni_x_Mn_14−x_ system [29]. According to their research, the authors revealed that martensitic Fe_86_Ni_x_Mn_14−x_ compounds have a lower transformation temperature than conventionally prepared alloys. Nonetheless, little attention has been dedicated to the bulk alloy’s magnetic behaviors and mechanical properties of the ternary. According to the literature [30,31,32], the most common methods of preparing bulk alloys are arc melting and powder metallurgy. In general, the microstructure of arc-melted alloys is the typical dendritic structure. Post-treatment is always used to remove composition segregation and other defects for improved properties. Mechanically alloyed powders can be rapidly consolidated in a short time by SPS [33,34,35]. Consequently, MA and SPS are now widely used for preparing bulk alloys [36]. Mechanical alloying can achieve grain refinement and homogeneous composition [37], while SPS allows for the production of fully dense stable solid solutions.

In the current work, a Fe_65_Ni_28_Mn_7_ alloy (low Mn content) is selected to support and improve the alloy properties based on a trade-off of (a) high strength, (b) good hardness, and (c) good magnetic behavior. Arc melting (AM), mechanical alloying (MA), and MA followed by spark plasma sintering (SPS) were used to produce these alloys, and they will be compared. The influence of milling time (MA) and the elaboration process (AM or SPS) on the alloy’s morphology, phase composition, structure, and microstructure are examined. The magnetic characteristics of as-milled powder, as-cast specimens, and MA + SPS samples were investigated, as well as the mechanical properties of the samples obtained after SPS.

## 2. Materials and Methods

Fe_65_Ni_28_Mn_7_ was prepared by the MA process from pure elemental powders of iron (purity 99.9%, <10 μm, spherical, Alfa Aesar, Haverhill, MA, USA), nickel (purity 99.9%, 3–7 μm, Strem, Newburyport, MA, USA), and manganese (purity 99.95%, <10 μm, Alfa Aesar). A high-energy planetary ball mill (Type Fritsch P7, Fritsch, Idar-Oberstein, Germany) with a ball-to-powder weight ratio of approximately 4:1 was used at a speed of 600 rpm in an argon atmosphere and at room temperature. To prevent excessive heating, sticking of the powder to the container walls and the balls, and powder agglomeration, the milling sequence was designed as 10 min of milling followed by 5 min of break.

The phase identification and the structural evolution were investigated using X-ray diffractometry (Bruker, Billerica, MA, D-500 S, USA) with CuKα radiation. The structural parameters were determined from the Rietveld refinement of the XRD patterns using the Maud 2.9 software [38]. The morphology of the milled powder was examined using scanning electron microscopy (SEM) (Carl Zeiss GmbH, DSM960A, Jena, Germany) in secondary electron mode operating at a voltage of 15 kV. The SEM was equipped with a Vega©Tescan (Tescan, Brno, Czech Republic) energy dispersive X-ray spectrometry (EDS) analyzer.

A superconducting quantum device with a maximum applied field of 20 kOe from Quantum Design ((Caledonia, MI, USA) SQUID MPMS-XL) was used at 300 K to measure the coercive field (Hc) and saturation magnetization (Ms) of the as-milled powders. The milled powders were then sintered by SPS (HPD10, FCT Systeme GmbH, Frankenblick, Germany) in a vacuum environment.

The consolidation was performed at 750 and 1000 °C under two pressures of 50 and 75 MPa at a heating rate of 50 °C/min. Then, the sample disc (10 mm diameter and 3 mm thick) cooled to room temperature within 10 min. In parallel, an as-cast ingot of about 4 g with nominal composition Fe_65_Ni_28_Mn_7_ was prepared by conventional arc melting (Type Buhler MAM-1 compact arc Melter) in Ar atmosphere from pieces of individual elements (99.98% pure Fe, 99.98% pure Ni, and 99.98% pure Mn). The ingot was melted five times and cast into a chilled copper mold to obtain a master rod with a diameter of 20 mm. After the polishing process, the densities of the “as-sintered” by spark plasma sintering (SPS) and “as-cast” by arc melting (AM) samples were determined using Archimedes’ method. The SPS and the AM ingot samples were also characterized by SEM and XRD, and their magnetic properties were measured. The Vickers hardness measurements of the obtained specimens were performed under a load of 200 g applied for 10 s. To calculate the average hardness, nine indentations were made. Further, the compressive characteristics were performed using a SHIMADZU AGX-V (Shimadzu, Kyoto, Japan) testing machine. The compression tests were performed on the solid samples acquired after AM and SPS after being cut into rectangular parallelepipeds, with a speed of 0.1 mm/min. Following the compression tests, the corresponding fracture surface characteristics were examined using SEM.

## 3. Results

### 3.1. Mechanical Alloying

#### 3.1.1. Morphological Changes

Figure 1 shows the morphological evolution of Fe_65_Ni_28_Mn_7_ alloy powders after various milling times. According to Figure 1a, typical mixture powder particle morphologies range from spherical (the majority) to irregular (the minority). Thus, there is no apparent deformation involved in the mixing process. A phenomenon of cold welding of the small particles on the surface of the big ones is seen after 4 h of milling, which can favor the growth of some particles in comparison to the smallest ones (Figure 1b) [39]. During this period of milling, agglomeration takes place, and the particle shape remains spherical.

The size distribution of powder particles appeared over a wide range (2–8 μm). Between 10 and 85 h of milling, the particles were discovered to be irregular in shape and flattened due to the continual severe plastic deformation that occurred along with cold particle welding during this period of the milling (Figure 1c–e). At this period of milling, the particle sizes undergo a considerable increase up to a value of 40 ± 7 μm. By extending the milling period to 130 h, the particle size steadily decreased to 20 ± 5 μm and the particle shape became roughly spherical due to the powder hardening and fracture. This is the result of the repeated ball shocks. It is the outcome of a severe fracture period (Figure 1f). Under the influence of extreme plastic deformation, the balance between fracturing and welding appears to be influenced by the alloy composition and milling duration. Early in the milling process, metallic particles are brittle and have a relatively high tendency to fuse together to produce larger particles. By extending the milling process, the tendency to fracture and work hardening of the particles predominates over cold welding, and the particle size decreases.

#### 3.1.2. Structural Changes

Figure 2 shows the XRD patterns of Fe_65_Ni_28_Mn_7_ powders at various milling times. Before milling, the recorded peaks correspond to the free BCC-Fe, FCC-Ni, and BCC-Mn elements constituting the powder mixture. After 10 h of milling, both (a) a slight broadening of the XRD peaks and (b) a significant decrease in their intensity is observed. This is mainly due to the refinement of the size of the grains and the crystallites, accompanied by an increase in the rate of deformation of the lattice and defect density (stacking faults, dislocations, grain boundaries, etc.) [40,41]. Inter-diffusion between Fe, Ni, and Mn atoms is excluded in this grinding period because neither shift nor asymmetry of diffraction peaks is noted. Raanaei et al. [42] and Luo et al. [43] had previously reported that there was no inter-diffusion between Fe and Ni between 16 and 10 h of milling. The decrease in the Mn peak located at 2-Theta = 43° occurred as the milling duration increased. This Mn peak seemed to vanish after 25 h of milling. Generally, the disappearance of XRD pattern peaks can be attributed to the formation of a solid solution, lattice distortion, or crystal refinement [44,45,46].

The XRD patterns were analyzed via the Rietveld technique using MAUD 2.9 software. An example of Rietveld refinement is given in Figure 3 for four experimental XRD patterns: un-milled powders and those milled for 25, 50, and 130 h. The goodness of fit (GOF) values of the refinement mentioned in Figure 3 show that the refinement can be trusted. The Rietveld analysis allowed us to obtain the phases proportion, lattice parameters, size of the crystallites, microstrain, and their error bars.

Figure 4a shows the monitoring of the evolution of the profile of the peak located at position 43.5°, specific to (111)_FCC_ and (110)_BCC_, as a function of milling time. This allows us to fit and analyze peak asymmetry and shift during the milling process. From 25 h of milling, the result of the fitting procedure shows that this large peak appears asymmetric due to the overlapping of the two peaks specific to two phases of different crystalline structures: the first one is located at a position of 2 theta lower and is specific to the FCC Ni-rich phase, whereas the second is placed 2 Theta higher and results from the presence of the BCC Fe-rich phase. By increasing the milling time up to 130 h, the proportion of the FCC phase increases in favor of the BCC phase (Figure 4b). At the end of milling, the proportion of the FCC phase reaches a value of about 76%, while that of the BCC phase becomes about 24%.

A more detailed examination of the XRD findings reveals that the XRD peaks broaden gradually as the MA process progresses. This could be caused by a reduction in crystallite size and an increase in lattice strain due to the accumulation of structural defects such as stacking faults, twins, dislocations, and grain boundaries [47]. Figure 5 and Figure 6 show the dependence of the calculated crystallite size and lattice strains, obtained by Rietveld refinement, on the milling time.

The refinement of the microstructure during mechanical alloying is significant, and it is in the order of a few nanometers. Increasing the milling time reduces the crystallite sizes and enlarging the lattice strains, first for the Fe, Ni, and Mn elemental powders (Figure 5) and then of the FCC-Ni-rich and BCC-Fe-rich solid solutions (Figure 6). The crystallite sizes for the BCC and FCC solid solutions decrease to approximately 9 and 11 nm, respectively. In parallel, the lattice strains increase up to 1.2 and 0.9% for the BCC and FCC phases, respectively.

Figure 7 depicts the change in the lattice parameters as a function of the milling time. The incorporation of Mn atoms into the vacant sites of pure Fe extends its lattice parameter slightly during the first hours of milling. After 25 h of milling (dashed red line on the graphics), the BCC-Fe-rich phase and the FCC-Ni-rich phase, respectively, with lattice parameters a = 2.8668 (1) Å and a = 3.5638 (1) Å had formed. After 130 h of milling, as a result of the substitution of Fe and Ni atoms, their lattice parameters became a = 2.8674 (1) Å and a = 3.5844 (1) Å, respectively. The increase in the lattice parameter may be caused by an increase in the density of dislocations, with their characteristic strain fields on the nanograin boundary [48]. In addition, this increase in lattice parameters for both phases may also be related to the substitution of Fe and Ni by Mn. Indeed, the Mn atom has an atomic radius (R_Mn_ = 1.79 Å) different from that of the Fe (R_Fe_ = 1.72 Å) and Ni (R_Ni_ = 1.62 Å) atoms, where the atomic radius ratios are R_Fe_/R_Mn_ = 0.96 and R_Ni_/R_Mn_ = 0.90.

Regarding the crystalline size, a diminution in size is found in the BCC phase from 26 nm at 25 h of milling to a value close to 9 nm at 130 h of milling. In the FCC phase, the crystalline size remains stable in the range of 12–15 nm (12 nm at 130 h). Thus, the final crystalline size of both phases is similar. The microstrain increases with milling time to 1.2% in BCC phase and 0.9% in FCC phase after milling for 130 h.

The formation and the mobility of dislocations are frequently cited as the origin of the lattice strain brought on by MA [49]. According to Fecht [50], grain size could be reduced through the formation and movement of dislocations. Rawers and Cook [51] showed that the strain on the nanograin boundary could extend into the nanograin, expanding the lattice. Therefore, dislocations are the major defects for MA materials that have received substantial plastic deformation, and this dislocation density, *ρ_D_*, may be expressed in terms of *D* and *ε* by the relation [52,53,54]:(1)$$\rho_{D}=2\sqrt{3\;}\frac{{<\varepsilon^{2}>}^{1/2}}{Db}$$
where *b* is the magnitude/module of the Burgers vector, *D* is the crystallite size, and *ε* is the lattice strain. The development and slipping of dislocations on <111> close-packed direction cause cold plastic deformation in BCC materials [52]. In terms of FCC materials, the close-packed direction is <110> [53]. The Burgers vector describes the magnitude and the direction of lattice distortions, and its intensity is defined by [55]:(2)$$b=\frac{a}{2}\sqrt{h^{2}+k^{2}+l^{2}}$$
where *h, k*, and *l* are the Miller indices, and a represents the unit cell length. As a result, Burger’s vectors for BCC and FCC phases are (*a*√3)/2 and (*a*√2)/2, respectively.

Figure 8 shows the calculated BCC-Fe-rich and FCC-Ni-rich phases dislocation densities in the mechanical alloy Fe_65_Ni_28_Mn_7_. For the BCC-Fe-rich phase, we can see a significant increase in *ρ_D_* from about 0.004 × 10^16^ m^−2^ to 0.534 × 10^16^ m^−2^ as the milling time increases from 1 to 25 h, and then it remains unchanged when increasing milling time up to 130 h, whereas for the FCC-Ni-rich phase, the dislocation density increases from about 0.001 × 10^16^ m^−2^ at 1 h to 0.55 × 10^16^ m^−2^ at 85 h. For extended times, a stabilization step is also recorded.

### 3.2. Arc Melting

Figure 9 illustrates the XRD pattern, SEM micrograph in secondary electron (SE) mode of the core, and the EDS results of the specimen after Arc melting. The sample’s XRD pattern shows that the alloy has a single FCC-Ni-rich phase. This finding implies that annealing results in a stable FCC phase at high temperatures, probably favored by a higher FCC phase content. For SEM analysis, the sample was mechanically polished up to a colloidal silica suspension of 50 nm. The super-finishing stage was extended to 20 min. As shown in the SEM micrograph, the as-cast sample shows a density of 96.22%, and a porosity of about 3.8% seems to be present. The deduced density value for this sample is 7.870 g/cm^3^. Unfortunately, the presence of pores tends to reduce the mechanical and fatigue characteristics of the composites. The presence of pores can be explained by relatively fast cooling during the peritectic and eutectic reactions that occur during the sample’s melt and remelt. The sample’s corresponding energy-dispersive X-ray spectra (EDS) show that the atomic concentrations of Fe, Ni, and Mn are 65.9, 28.8, and 5.3 at. %, respectively, which are very close to the nominal composition of Fe_65_Ni_28_Mn_7_.

### 3.3. Spark Plasma Sintering

After 130 h of milling, the Fe_65_Ni_28_Mn_7_ powders are consolidated into bulk samples by SPS at 750 °C/50 MPa, 750 °C/75 MPa, and 1000 °C/50 MPa. Their corresponding XRD patterns and SEM images are given in Figure 10. The deduced density values for these samples are 7.563 g/cm^3^ for MA + SPS at 750 °C/50 MPa, 7.673 g/cm^3^ for MA + SPS at 750 °C/75 MPa, and 7.680 g/cm^3^ for MA + SPS at 1000 °C/50 MPa. The XRD pattern of the bulk samples processed by SPS at 750 °C/50 MPa shows the coexistence of the FCC and BCC phases, and the BCC phase exists in a small proportion (20%). By performing SPS at a high temperature, the grains increased and internal tensions were released, sharpening the XRD peaks. According to the Rietveld analysis, the size of the crystallites is estimated to be around 50 nm. In addition, we notice well-defined diffraction peaks at the 2θ values of 35.12, 40.64, 58.55, and 70.14°, which are indexed to (311), (511), (440), and (444) crystal plane reflections of spinel-type MnFe_2_O_4_ (JCPDS 73-1964), respectively. As shown in the SEM image, one can observe good densification of the milled powder particles where the relative value of the density is estimated to be 97.66% and the porosity is about 2.34%. When the pressure is raised to 75 MPa and the temperature remains at 750 °C, one can observe that the BCC phase completely vanishes, suggesting that the total disappearance of the BCC phase when pressure is raised is due to the phase transition from BCC to FCC [56,57,58]. In addition, the pressure causes the density to rise to 97.76% and the porosity to fall to 2.24%. The size of the pores shrank when the pressure was raised to 75 MPa, but they are still discernible. An increase in the crystallite size of the FCC phase was noted and calculated to be about 47 nm. On the other hand, further increasing the sintering temperature to 1000 °C under 50 MPa pressure causes the FCC phase’s crystallite size to expand to more than 100 nm, its density to slightly increase to 98.54%, and its porosity to significantly reduce by 1.50%. A high annealing process is required for the good consolidation of powders to bulk material, that is, to remove essentially all porosity and to achieve good interparticle bonding. By comparing the crystallite size results obtained after the three metallurgical processes, mechanosynthesis, arc melting, and SPS processes, one can conclude that by SPS, the crystallite growth is limited, and the resulting nanostructure after this process is maintained [59]. As shown in Figure 10, the oxide inclusions and FCC phase grains made up the microstructure. When metallic powders are exposed to air, oxide shells form on their surfaces. The milling process repeatedly fractures and cold re-welds the original elemental powder particles, causing the oxides at the particle surface to generate a homogenous dispersion in the cross-section of the individual particles. Focusing on the SEM pictures, one can observe that some of the oxides were grouped as strings in addition to individual particles. Thus, the oxides present on the original surfaces of the as-milled powder particles prior to the SPS process are most likely to be responsible for this characteristic. During high-temperature sintering, the oxides shift form to reduce surface energy (naturally achieving a spherical shape).

Another feature in the XRD patterns is that all the peaks are shifted to little values of angles in the case of the samples processed at 750 °C under 75 MPa and 100 °C under 50 MPa. This is due to the increase in the lattice parameters during the SPS process. The maximum variations in the crystal parameter were 0.0022 and 0.0025 Å for the samples obtained by SPS at 750 °C/75 MPa and 1000 °C/50 MPa, respectively. This increase in the lattice parameter is probably explained by the dissolution of the BCC phase in terms of Fe in the Ni matrix. Indeed, the Fe atom has an atomic radius (R_Fe_ = 1.72 Å) different from that of the Ni (R_Ni_ = 1.62 Å) atoms. This effect of the atomic radius may be added to that of the plastic deformation following the SPS process, which can also give an expansion of the crystal lattice.

### 3.4. Magnetic Behavior

Figure 11 depicts the hysteresis loops (M-H) of mechanical alloying powder, arc melting specimens, and spark plasma sintering samples processed at 750 °C and 1000 °C. All of the samples displayed comparable hysteresis loops, which is characteristic of a ferromagnetic state. However, the change in magnetic properties can be related to the microstructural changes in the sample after each metallurgical process, as measured by the XRD technique. The determination of the characteristics of the magnetic parameters can be accomplished through the use of M-H curves. Among these parameters, the coercive field (Hc) is determined. Generally, the Hc of crystalline alloys is heavily dependent on the microstructure evolution of metallic materials throughout the milling process thanks to the interaction of magnetic domain walls with the grain boundaries. On the other hand, nanocrystalline ferromagnetic materials offer new possibilities for tailoring advantageously a variety of phenomena, including soft (Hc < 1000 A/m (12.56 Oe)), semi-hard (10,000 A/m > Hc > 1000 A/m), hard (Hc > 10,000 A/m), and superparamagnetic (Hc~0 A/m) behaviors [60].

Table 1 shows the saturation magnetization (Ms), coercivity (Hc), and remanence to saturation ratio (Mr/Ms) values. As shown, the specimen obtained by mechanical milling exhibited a hard magnetic behavior, where Hc equals 24.7 Oe, corresponding to a semi-hard behavior. It is worth noting that the coercive field of the as-milled powder is high when compared to pure iron (796 A/m^−10^ Oe) and pure nickel (557 A/m^−7^ Oe) [61,62]. However, both sintered samples at 750 °C under a pressure of 50 MPa and 75 MPa have values of coercivity of 0.60 Oe. Increasing the sintering temperature to 1000 °C under a pressure of 50 MPa resulted in an improvement in coercivity to 0.07 Oe. As a result, these three sintered samples can be considered alloys with soft magnetic properties. The decrease in the value of Hc with an increasing sintering temperature may be due to the release of structural strain created during the MA process. As a result of the FCC-rich Ni phase’s presence at the expense of the BCC-rich Fe phase, sintered samples are magnetically soft, due to the fact that a hard–soft ferromagnetic transition occurs as a result of temperature evolution. According to the random anisotropy model [63], soft magnetic materials experience an unusual magnetic hardening when the grain size is greater than or equal to the ferromagnetic exchange length, L_ex_. The estimated value of L_ex_ for Fe-based alloys is in the range of 10–20 nm, which roughly corresponds to the D value of the FeNiMn alloy (~10 nm) obtained after 130 h of milling. In the current investigation, arc melting and sintering processes result in a reduction in Hc when the L_ex_ is higher than D. On the other hand, the as-milled powder had a magnetization saturation of around 122 (emu/g). The magnetic saturation of the SPS at 750 °C/75 MPa is lower than the magnetic saturation of the SPS at 750 °C/50 MPa. This finding might be explained by the removal of the BCC phase when the pressure is increased. Previous studies have indicated that the FCC phases increase magnetic saturation [64]. Increasing the sintering temperature to 1000 °C under a pressure of 50 MPa resulted in an improvement in saturation magnetization to 118.10 (emu/g). Similar research has found that increasing the sintering temperature causes a rise in saturation magnetization [65]. The saturation ratio increased from 0.020 after milling to 0.11 after SPS at 750 °C under a pressure of 50 MPa and to 0.10 after SPS at 750 °C under a pressure of 75 MPa, then declined to 0.074 after SPS at 1000 °C under a pressure of 50 MPa. This drop could be attributed to the release of microstrain and domain wall energy as the temperature rises. After SPS at 1000 °C, it is possible to conclude that a good balance of saturation magnetization, coercivity, and the saturation ratio is reached. As a result, as the ferromagnetic material’s temperature rises, the thermal vibrations of its atoms also increase. The magnetic moments then become freer to rotate and are subsequently ordered randomly. The temperature has an impact on magnetic properties because of this, considerably and quantitatively decreasing them.

### 3.5. Mechanical Properties

The Vickers hardness (HV), compressive strength (σmax), yield strength (σy), and shortening at failure values are listed in Table 2. The compression tests were made on small parallelepipeds. All the SPS samples have significantly higher HV, yield strength, and compressive strength than the AM sample, which is related to the smaller grain size, higher density of dislocations, and presence of dispersed oxides within the material. These results are consistent with those reported by Yang et al. [65]. The HV, σy, and σmax, on the other hand, are highly dependent on processing technology. In other words, they depend on dislocation strengthening and ultrafine grains. However, as with the AM sample, increasing the temperature generates an increase in grain size. According to the Hall–Petch relationship, increasing grain size reduces compression yield strength significantly [66,67]. By comparing the samples obtained by spark plasma sintering, we can conclude that the sample sintered at 750 °C under 75 MPa is significantly harder (425 HV) than the sample sintered at the same temperature under a lower pressure of about 50 MPa (due to the higher density of the dislocations) and the sample sintered at 1000 °C (due to smaller grain size). It also has a higher yield strength of 1350 MPa and compressive strength of 1700 MPa. As expected, the arc-melted alloy shows the lowest hardness, yield strength, and compression strength. It is worth noting that the samples sintered at 750 °C had a higher hardness than the Hadfield Fe-Mn steel (~248 HV) [68]. The high yield strength is an interesting characteristic. It is often accompanied by higher resistance to wear and lower ductility. All the samples consolidated by SPS showed very interesting shortening at failure values A% superior to 15% and brutal noisy breaking during the compression tests, while the arc melted sample supported shortening superior to 30% (test stopped at 30%, no failure).

Figure 12 compares the SEM images of the fracture surfaces of the specimens produced by arc melting and spark plasma sintering after compression tests. It is important to note that the specimen synthesized by MA + SPS at 1000 °C/50 MPa did not break during the compression test, indicating that it is a ductile material. The image of the surface of the sample MA + SPS 1000 °C/50 MPa is presented in Figure 12d. The surface of the specimen formed via Arc melting (not fractured, c.f. Figure 12a) exhibits a typical surface of a homogeneous ductile material. One can clearly see different slip planes in different grains, with different orientations, which gives at the free surface step-like protrusions. Large slip lines can be seen inside, as well as smaller ones arranged in chevrons. This is indicative of a very homogenous material that has been exposed to significant deformation. In the case of the specimen produced by SPS at 750 °C/50 MPa (Figure 12b), the grains can be seen on the fracture surface. It could be a shearing morphology provoked by the compressive stress or an indication that the rupture was intergranular. Moreover, the visible dimples indicate that the rupture was primarily ductile. Increased sintering pressure to 75 MPa (Figure 12c) leads to a notable change in fracture mode to transgranular mode due to the strengthening of powder particle bonds, and the absence of dimples suggests that the fracture was brittle. The fracture brittleness may be predicted from the Vickers hardness, which shows that the sample sintered at 750 °C under 75 MPa is considerably harder (425 HV). The high-resolution images presented in Figure 13 show very different materials.

In the case of the SPS 750 °C/50 MPa (Figure 13a), the fracture is mostly intergranular, while the SPS 740 °C/75 MPa (Figure 13b) shows a clearly transgranular fracture. Interesting features are observed on the SPS 750 °C/50 MPa image, with agglomerates (the result of cold-welded nanograins after MA) still clearly identifiable. Some porosity can be remarked too, already seen in Figure 10 in the cross-section SEM images of the unfractured samples. On the fractographic image of the SPS 750 °C/75 MPa, no pores are visible, which suggests perfectly welded grains.

## 4. Conclusions

A comparison study of the microstructures, phase evolution, magnetic behavior, and mechanical properties of the Fe_65_Ni_28_Mn_7_ alloy prepared by mechanical alloying (MA), arc melting (AM), and spark plasma sintering (MA + SPS) was conducted, and the following conclusions were reached:Mechanical alloying results in the formation of two supersaturated solid solutions after 130 h of milling, a BCC phase with a crystallite size of 9 nm and a major FCC phase with a crystallite size of 12 nm.Arc melting results in the formation of a single FCC phase.Spark plasma sintering after the mechanical alloying (MA + SPS) revealed the disappearance of the BCC phase. The sintered alloys also showed an FCC phase with a crystallite size in the nanometer scale < 95 nm depending on the sintering process.The powder obtained after MA, as well as all sintered alloys, exhibits soft magnetic behavior. The alloy sintered at 1000 °C under 50 MPa demonstrated soft magnetic properties, saturation magnetization (Ms) of 118.10 emu/g, and coercivity (Hc) of 0.07 Oe, qualifying it as a suitable candidate for soft magnetic applications, while the alloy obtained by AM demonstrated a hard magnetic behavior.Sintering alloys have demonstrated excellent mechanical properties, with Vickers hardness, compressive yield strength, and shortening at break (compression tests) values exceeding 216 HV, 1400 MPa, and 15%, respectively. The alloy sintered at 750 °C under 75 MPa demonstrated the best mechanical combination, with Vickers hardness, yield strength, and shortening at break values of 425 HV, 1700 MPa, and 16%, respectively. The alloy obtained by AM presents lower values of hardness and yield strength but high ductility, with a value of strain to failure of about 30%.The alloys produced by arc melting and MA + SPS at 1000 °C/50 MPa are ductile materials. However, sintering the powder at 750 °C and increasing the pressure from 50 to 75 MPa causes the fracture to change from intergranular to transgranular.

In light of the above observations, it is reasonable to conclude that the Fe_65_Ni_28_Mn_7_ alloy made by (MA + SPS) is the alloy that has the most promise, as it can be sintered at 1000 °C/50 MPa to increase magnetic softness and 750 °C/75 MPa to improve mechanical characteristics. For usage in mechanical or magnetic applications, sintered alloys might be a fantastic option. Attractive compounds for industrial applications should result from further work on the optimization of SPS conditions. If the SPS conditions are optimized further, the current sintered process following mechanical alloying can be an ideal option for industrial applications.

## Figures and Tables

**Figure 1 materials-16-07244-f001:**
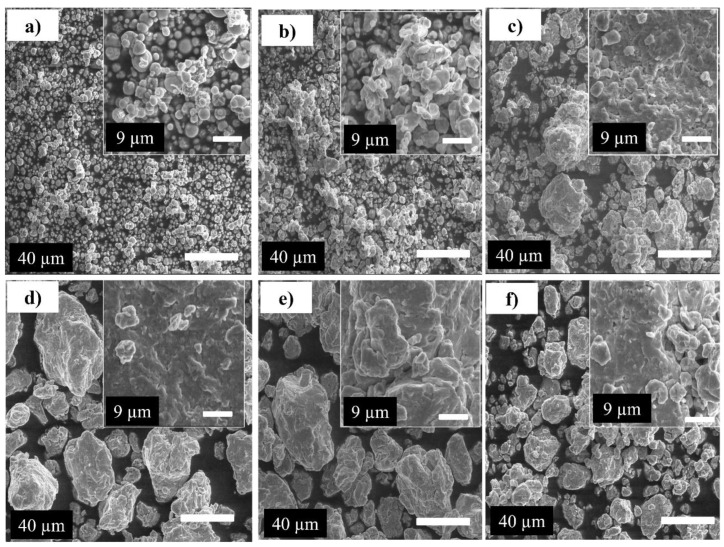
Scanning electron micrographs corresponding to mechanically milled powders: (**a**) 0 h, (**b**) 4 h, (**c**) 10 h, (**d**) 25 h, (**e**) 85 h, and (**f**) 130 h.

**Figure 2 materials-16-07244-f002:**
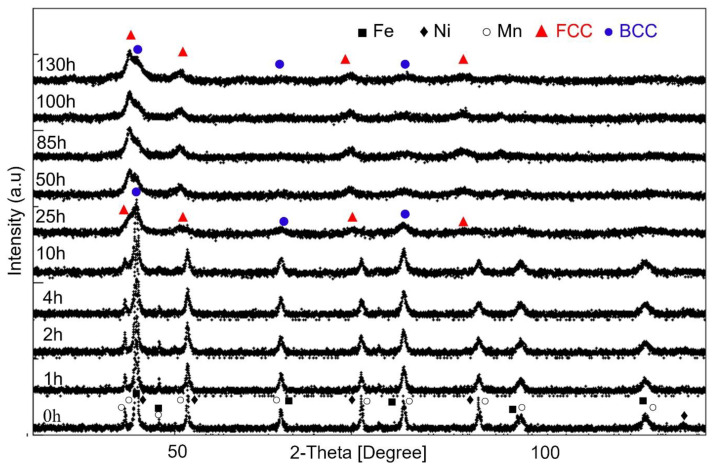
X-ray diffraction patterns of Fe_65_Ni_28_Mn_7_ powders as a function of mechanical alloying time.

**Figure 3 materials-16-07244-f003:**
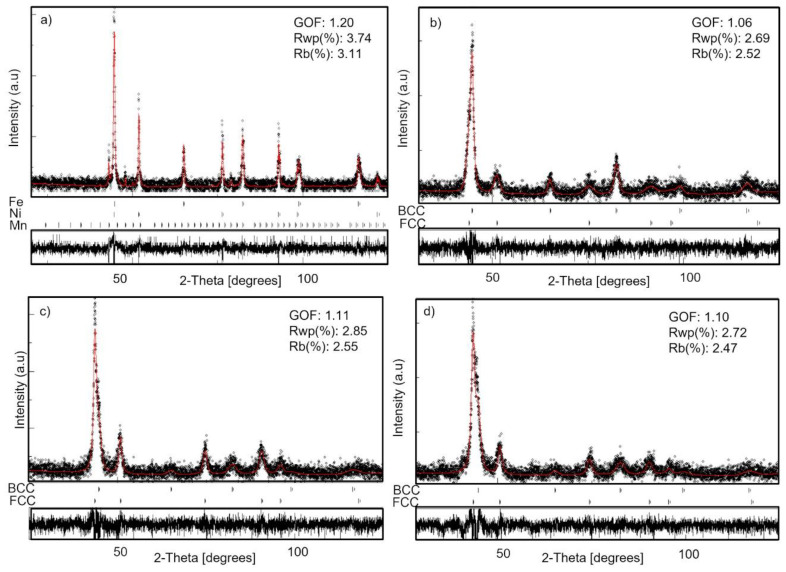
Rietveld refinement of the XRD patterns of the Fe_65_Ni_28_Mn_7_ powder: (**a**) before milling, (**b**) 25 h, (**c**) 50 h, (**d**) 130 h (scatters: experimental data, red solid line: fit).

**Figure 4 materials-16-07244-f004:**
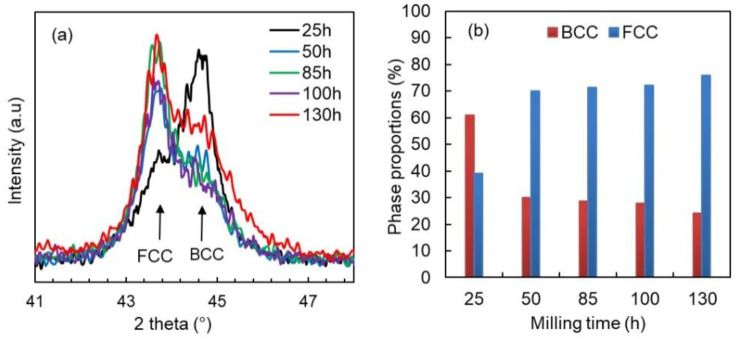
Evolution during the milling of the mains BCC and FCC XRD peaks (**a**) and phase proportions (**b**).

**Figure 5 materials-16-07244-f005:**
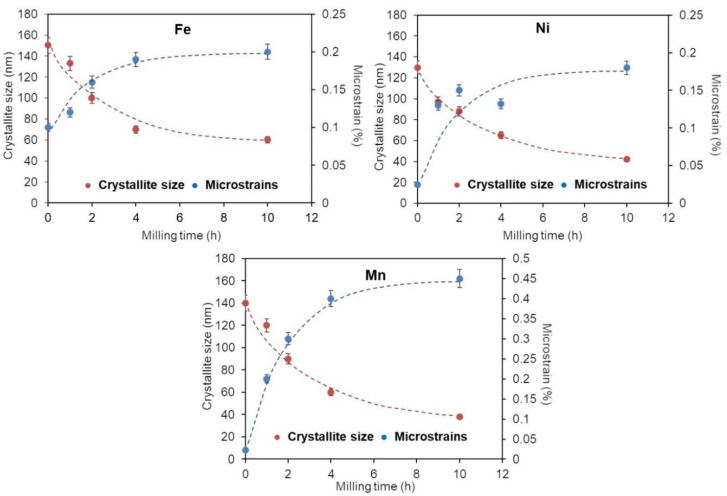
Evolution of the microstructure parameters of the Fe, Ni, and Mn elemental powders during the milling.

**Figure 6 materials-16-07244-f006:**
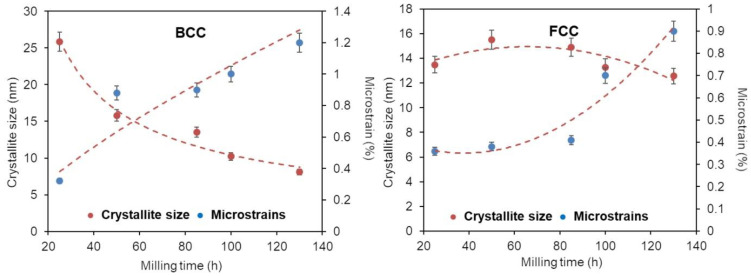
Evolution of the microstructure parameters of the BCC-Fe-rich and FCC-Ni-rich phases during the milling times.

**Figure 7 materials-16-07244-f007:**
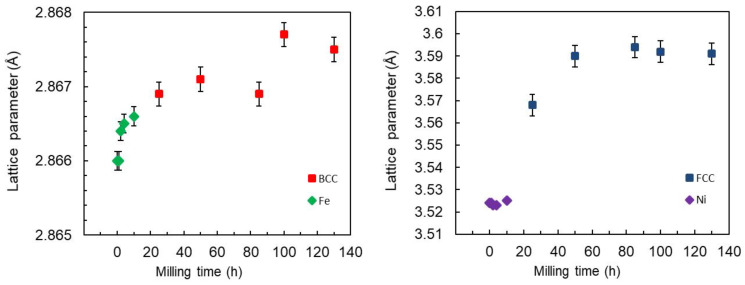
Evolution of the lattice parameters of the Fe, Ni, BCC-Fe-rich, and FCC-Ni-rich phases during the milling times.

**Figure 8 materials-16-07244-f008:**
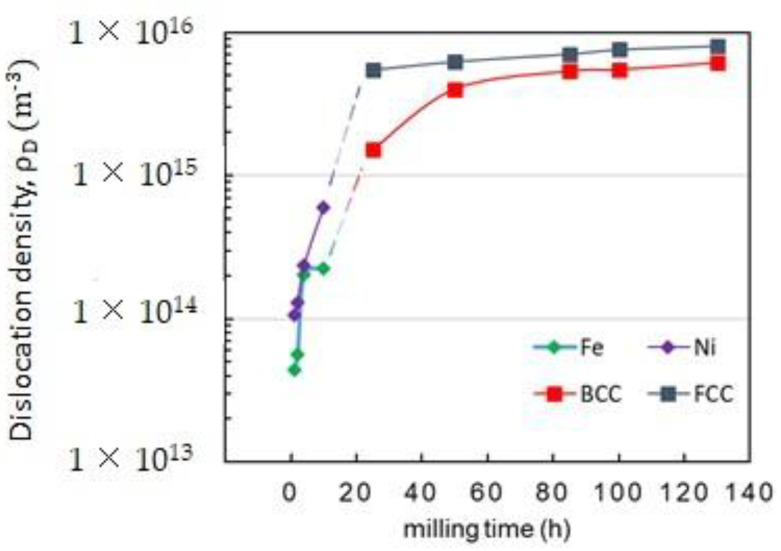
Variation of dislocation densities of the Fe, Ni, BCC-Fe-rich, and FCC-Ni-rich phases during the milling times.

**Figure 9 materials-16-07244-f009:**
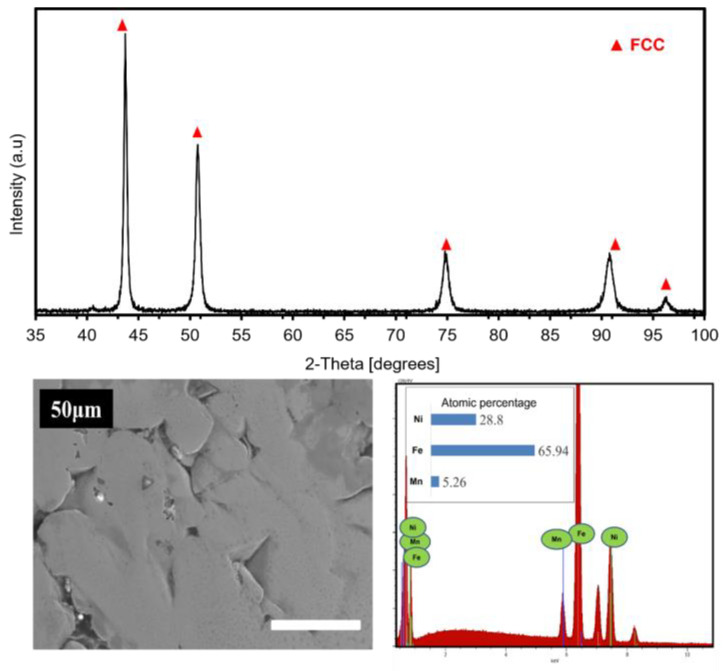
XRD patterns, SEM image, and corresponding EDS (counts versus energy, lines: blue Mn, red Fe, green Ni) analyses of the Fe_65_Ni_28_Mn_7_ alloy synthesized by Arc melting process. The EDS calculated percentages are: 28.8% Ni, 65.94% Fe, 5.26% Mn.

**Figure 10 materials-16-07244-f010:**
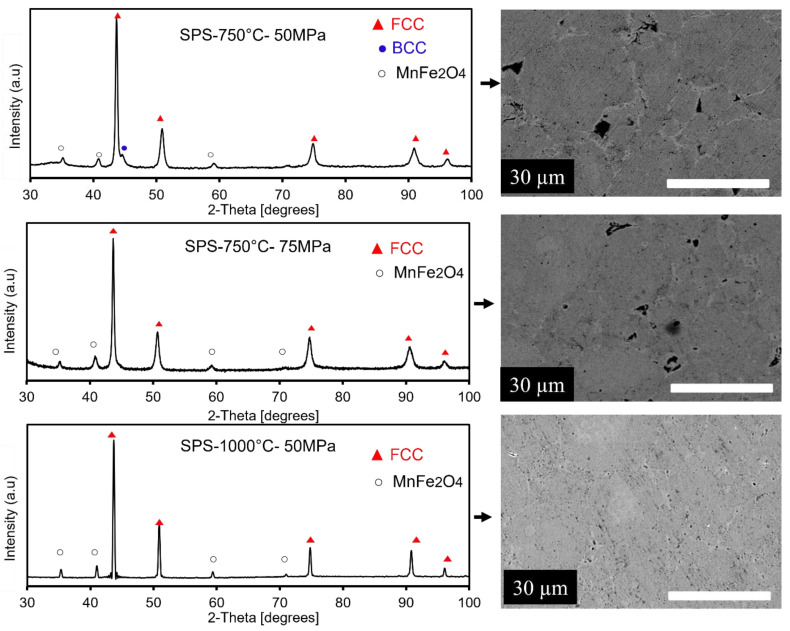
XRD patterns and the scanning electron micrographs of the Fe_65_Ni_28_Mn_7_ alloy after SPS.

**Figure 11 materials-16-07244-f011:**
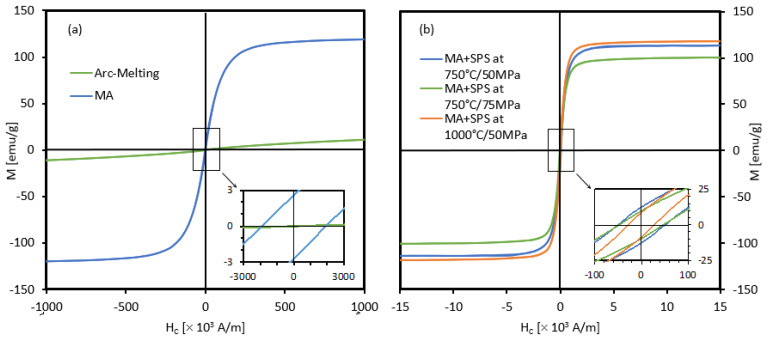
Magnetic hysteresis plots of the Fe_65_Ni_28_Mn_7_ alloy after three processes, (**a**) MA and Arc melting and (**b**) SPS at 750 °C/50 MPa, 750 °C/75 MPa, and 1000 °C/50 MPa. The box indicates the enlarged area in the insert.

**Figure 12 materials-16-07244-f012:**
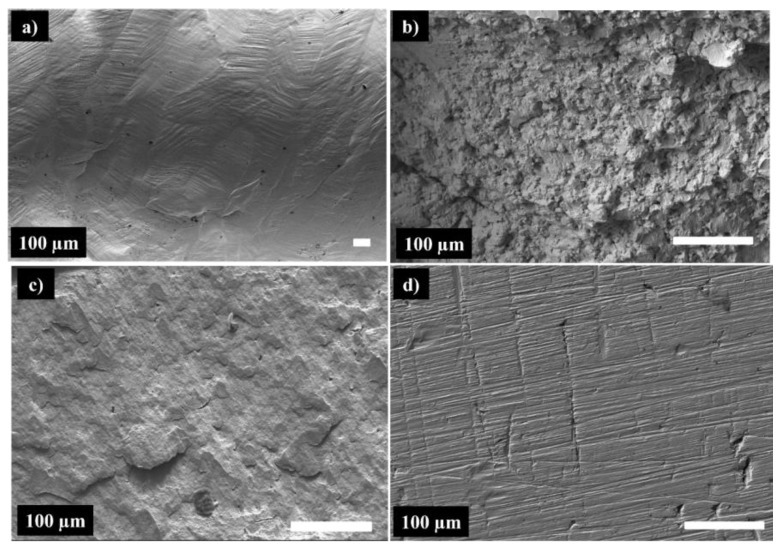
SEM image of the deformed surface of the Arc melting sample (**a**), fractography of the specimens synthesized by MA + SPS at 750 °C/50 MPa (**b**), fractography of the specimens synthesized by MA + SPS at 750 °C/75 MPa (**c**), and SEM image of the deformed surface after MA + SPS at 1000 °C/50 MPa (**d**).

**Figure 13 materials-16-07244-f013:**
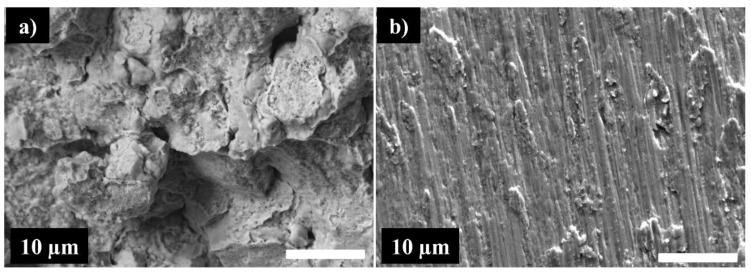
High-resolution fractographic images of the SPS 750 °C/50 MPa (**a**) and SPS 750 °C/75 MPa (**b**) samples.

**Table 1 materials-16-07244-t001:** Magnetic characteristics and microstructure parameters of the Fe_65_Ni_28_Mn_7_ alloy after mechanical alloying, arc melting, and spark plasma sintering.

Samples	M_s_ (emug^−1^)	H_c_(Oe)	M_r_/M_s_	<D> (nm)	ε (%)	a (Å)	ρ_D_ (m^−2^)
As-milled	122.10	24.07	0.0209	12 (FCC) 8 (BCC)	0.9 (FCC) 1.2 (BCC)	3.591 (FCC) 2.867 (BCC)	0.810^16^ (FCC) 0.6110^16^ (BCC)
Arc Melting	18.81	2.26	0.0008	>100	-	3.591	<10^14^
MA + SPS at 750 °C/50 MPa	113.61	0.60	0.11	50.27	0.2	3.590	3.957 · 10^14^
MA + SPS at 750 °C/75 MPa	100.58	0.60	0.10	50	0.13	3.581	4 · 10^14^
MA + SPS at 1000 °C/50 MPa	118.15	0.07	0.074	>100	0.02710^−2^	3.591	<10^14^

**Table 2 materials-16-07244-t002:** Mechanical properties: compressive strength (σ_max_), yield strength (σ_y_), shortening at failure (A%), and Vickers hardness of the Fe_65_Ni_28_Mn_7_ alloy after arc melting and MA + SPS.

Samples	σ_y_ (MPa)	σ_max_ (MPa)	A%	Hardness (HV)
Arc Melting	123	>500 (no failure)	>30%	97
MA + SPS at 750 °C/50 MPa	1027	1470	15%	328
MA + SPS at 750 °C/75 MPa	1350	1700	16%	425
MA + SPS at 1000 °C/50 MPa	1056	>1440 (no failure)	>15%	216

## Data Availability

Data can be requested from the authors.

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
