# Peer review of "Study of Structural, Compression, and Soft Magnetic Properties of Fe65Ni28Mn7 Alloy Prepared by Arc Melting, Mechanical Alloying, and Spark Plasma Sintering"

_materials, 2023, doi:10.3390/ma16227244_

Round 1

Reviewer 1 Report

Comments and Suggestions for Authors

The structural, compressive, and soft magnetic properties of soft magnetic Fe65Ni28Mn7 (percent) alloy, materials produced by mechanical alloying and spark plasma sintering (SPS) were compared with the material produced by arc melting (AM) for the same composition comparison. SEM, XRD were used for characterization. Coercive field (Hc), and saturation magnetization (Ms) measurements were made. The study generally contains intense and detailed information. It may be of interest to readers in the field of interest. However, the questions/hesitations listed below are expected to be answered.

1.        The word "Bulk" in the title is inappropriate. Since the material used is in powder form, it refers to "Bulk" ingot material.

2.         Is it obvious that the selected "Fe65Ni28Mn7" material is soft magnetic? Was it specifically determined? Or were the soft magnetic properties examined according to the selected experimental parameters? For this reason, it is not fully understood what the main motivation of the study is in the Introduction. The "Introduction" section should be divided into paragraphs with appropriate topic sections. You must demonstrate the originality and innovative aspect of your work.

3.        In line 101, the initial particle sizes and properties of the iron, nickel and manganese powders used for Fe65Ni28Mn7 should be added.

4.        The abbreviations "as-sintered" (SPS) and "as-cast" (AM) should be used as abbreviations in line 126, and the expressions "as-sintered" and "as-cast" should be used as abbreviations, as in line 130.

5.        Have powder size analyzes been performed for the powder particles shown in Figure 1? It would be meaningful if information could be given about the size distribution histograms.

6.        6. It would be more appropriate to give the details of the calculations such as crystallite size, lattice strain, dislocation density, lattice parameters etc. mentioned in line 240 in the Experimental Studies section.

7.        7. In lines 245 and 246, the dense slip planes of the BCC and FCC crystal structures should be checked. It should be the other way around

8.        What is the relationship of the "milling time" parameter in the arc melting process in line 266?

9.        The result "... BCC phase with a crystallite size of 11 nm" on line 450 was not mentioned in the article. Check the items in the results section.

10.     In line 425, the expression "..the fracture is mostly intergranular, while the SPS 740 °C/75 MPa (Fig.13.b) shows a clearly transgranular fracture" is hesitant. It resembles the shearing morphology due to compressive stress in Figure 13 b.

11.     In line 134, information about the sample dimensions and test standard for the compression test should be given. Additionally, since the compression test was applied in this study, it would be more appropriate to add compression properties instead of mechanics in the article title.

12.     Since arc melting is applied in your work, this process should also be reflected in the title.

Author Response

  1. The word "Bulk" in the title is inappropriate. Since the material used is in powder form, it refers to "Bulk" ingot material.
    Answer: We delete the word “bulk” in the title.

  1. Is it obvious that the selected "Fe65Ni28Mn7" material is soft magnetic? Was it specifically determined? Or were the soft magnetic properties examined according to the selected experimental parameters? For this reason, it is not fully understood what the main motivation of the study is in the Introduction. The "Introduction" section should be divided into paragraphs with appropriate topic sections. You must demonstrate the originality and innovative aspect of your work.

Answer: We divided the introduction section: mechanical alloying, need of bulk specimens, magnetic behavior, mechanical behavior, …. We finish remarking the interest of our work. To produce Fe-Ni-Mn bulk specimens with optimizes mechanical and magnetic properties adding a small amount of Mn.

Mechanical alloying produces powdered samples. Thus, to produce piece is necessary the consolidation and/or sintering. Particularly with nanocrystalline soft magnetic materials, bulk product fabrication is complex. However, the large fabrication of bulk nanocrystalline material is required for commercial applications. The spark plasma sintering (SPS) process is currently increasing in popularity as a novel solidification casting technique [11-16]. By retaining the nanocrystalline or amorphous structure, the SPS approach may restrict grain growth during compaction and enable bulk material production. A suitable ferromagnetic substance is pure iron. It has an extremely low resistivity, which results in significant eddy current losses.

Regarding magnetic applications, devices utilizing alloyed iron have higher efficiency than those using pure iron cores because it has higher magnetic permeability and lower overall core losses. Further, different alloying components affect anisotropy and magnetization in different ways [17]. The solid solution's magnetic properties are supposed to change linearly with the weight percent added in the alloy if two components are assumed to be ferromagnetic and have known saturation magnetization values. Nickel metal is mixed with iron to increase electrical resistivity, which decreases eddy current loss. Additionally, higher permeability and saturation magnetization are produced by an increase in Ni content [18]. The Fe-Ni alloy combination therefore has good soft magnetic characteristics.

Regarding mechanical response, based on its influence on the strength of Fe grain boundaries, Seah [19] categorized the addition of several elements. According to the author, although Mo and C helped to strengthen the grain boundaries, components like P and Mn caused the borders to weaken. Squires and Wilson [20] hypothesized that embrittlement is caused by Mn atom segregation at grain boundaries during aging in light of the brittle fracture along primary austenite grain boundaries and the absence of second particles in the fracture surfaces. Heo [21] has also shown that irregular segregation of Ni and Mn atoms through grain boundaries in Fe-7Ni-8Mn wt.% alloy results in embrittlement that is consistent with the concept of heat embrittlement [19]. In addition, Steven and Balajiva [22] first described Mn as an element that causes brittleness in alloys. This idea was subsequently supported by Schultz and McMahon [23], and Weng and McMahon [24]. Further, Wilson [25] disagreed, however, that manganese segregates at grain boundaries as a first step in the production of grain boundary precipitates and functions as a significant embrittling component in the early phases of aging.

Therefore, the development of Fe-Ni-Mn bulk alloys preventing the brittleness is a field of scientific and technological interest. Thus, numerous attempts have been made on Fe–Ni–Mn alloys to address their brittleness problems, which can be arranged by the combination of mechanical alloying and SPS techniques. As known, Fe–Ni–Mn alloys are a class of high-strength coating martensitic alloys. They are sufficiently ductile in the solution-annealed state but suffer from severe intergranular embrittlement along preceding austenite grain boundaries [26, 27]. Also, martensitic transformation in Fe-Ni-Mn alloys has been widely investigated [28]. The mechanical alloying method was used to produce the Fe86NixMn14-x system [29]. According to their research, authors revealed the martensitic Fe86NixMn14-x compounds have a lower transformation temperature than conventionally prepared alloys. Nonetheless, little attention has been dedicated to the bulk alloy's magnetic behaviors and mechanical properties of the ternary. According to the literature [30-32], the most common methods of preparing bulk alloys are arc melting and powder metallurgy. In general, the microstructure of arc-melted alloys is the typical dendritic structure. Post-treatment is always used to remove composition segregation and other defects for improved properties. Mechanically alloyed powders can be rapidly consolidated in a short time by SPS [33-35]. Consequently, the MA and SPS are now widely used for preparing bulk alloys [36]. Mechanical alloying can achieve grain refinement and homogeneous composition [37], while the SPS allows the production of fully dense stable solid solutions.

Therefore, in the current work, a Fe65Ni28Mn7 alloy (low Mn content) is selected to support and improve the investigation based on a trade-off the combination of: a) high strength, b) good hardness, and c) good magnetic behavior. Arc melting (AM), mechanical alloying (MA), and MA followed by spark plasma sintering (SPS) were used to produce these alloys and these latter will be compared. The influence of milling time (MA) and elaboration process (AM or SPS) on the alloy's morphology, phase composition, structure, and microstructure are examined. The magnetic characteristics of as-milled powder, as-cast specimens, and MA+SPS samples were investigated, as well as the mechanical properties of the latter two.

  1. In line 101, the initial particle sizes and properties of the iron, nickel and manganese powders used for Fe65Ni28Mn7 should be added.

Answer: We add some information in the materials description.

Fe65Ni28Mn7 was prepared by MA process from pure elemental powders of iron (purity 99.9%, < 10 μm, spherical, Alfa Aesar), nickel (purity 99.9%, 3-7 μm, Strem), and manganese (purity 99.95%, < 10 μm, Alfa Aesar).

  1. The abbreviations "as-sintered" (SPS) and "as-cast" (AM) should be used as abbreviations in line 126, and the expressions "as-sintered" and "as-cast" should be used as abbreviations, as in line 130.

Answer: It is an option to clarify the manuscript. We modify using the SPS and AM abbreviations.

  1. Have powder size analyzes been performed for the powder particles shown in Figure 1? It would be meaningful if information could be given about the size distribution histograms.

Answer: There are some wide size distributions. Thus, we introduce a range or the mean value. Nevertheless, we add information about the standard deviation as a statistical value, excepting 0h because is a mixture of the original elemental particles.

  1. It would be more appropriate to give the details of the calculations such as crystallite size, lattice strain, dislocation density, lattice parameters etc. mentioned in line 240 in the Experimental Studies section

Answer: We introduce additional information in the manuscript.

Regarding the crystalline size, it is found a diminution in the BCC phase from 26 nm at 25h of milling to < 10 nm at 130 h. of milling In the FCC phase the crystalline size remain stable in the range 12-15 nm. Thus, the final crystalline size of both phases is similar. With respect the microstrain, it increases with milling time to 1.2% in BCC phase and 0.9% in FCC phase after milling for 130h.

The dislocations density information was given in a paragraph before figure 8.

  1. In lines 245 and 246, the dense slip planes of the BCC and FCC crystal structures should be checked. It should be the other way around

Answer: The slip planes of the BCC and FCC phases are well known in the scientific literature. Thus, this information providing of references. We remark this by remarking that this information is indicated in references [52] and [53].

  1. What is the relationship of the "milling time" parameter in the arc melting process in line 266?

Answer: We agree this comment. We modify this sentence.

This finding implies that annealing results in a stable FCC phase at high temperatures, probably favored by higher FCC phase content

  1. The result "... BCC phase with a crystallite size of 11 nm" on line 450 was not mentioned in the article. Check the items in the results section.

Answer:
We obtain this information from a figure. The value is 12 nm and we introduce in the results and discussion section.

Regarding the crystalline size, it is found a diminution in the BCC phase from 26 nm at 25h of milling of 9 nm at 130 h of milling. In the FCC phase the crystalline size remains stable in the range 12-15 nm (12 nm at 130 h). The microstrain increases with milling time to 1.2% in BCC phase and 0.9% in FCC phase after milling for 130h.

  1. In line 425, the expression "...the fracture is mostly intergranular, while the SPS 740 °C/75 MPa (Fig.13.b) shows a clearly transgranular fracture" is hesitant. It resembles the shearing morphology due to compressive stress in Figure 13 b.

Answer: We take into account the comment of the referee. We add this information in the manuscript.

It can be a shearing morphology provoked by the compressive stress or an indication that the rupture was intergranular.

  1. In line 134, information about the sample dimensions and test standard for the compression test should be given. Additionally, since the compression test was applied in this study, it would be more appropriate to add compression properties instead of mechanics in the article title.

Answer: We modify the title and we mentioned the shape of the samples during the compression test.

  1. Since arc melting is applied in your work, this process should also be reflected in the title.

Answer: We modify the title. We add “arc melting”.

Reviewer 2 Report

Comments and Suggestions for Authors

In the manuscript “study of structural, mechanical, and soft magnetic properties of bulk Fe65Ni28Mn7 alloy prepared by mechanical alloying and spark plasma sintering”, a variety of fabrication methods were employed to prepare bulk specimens composed of Fe65Ni28Mn7 (at.%). Different properties of the Fe65Ni28Mn7 samples fabricated by means of different methods were analyzed and compared. The current quality of the manuscript cannot meet the requirements for publication. In addition, the following questions should be taken into account.

1. The manuscript contains numerous writing mistakes and requires extensive revisions to improve its academic writing quality.

2. Line 46: The sentence “The spark plasma sintering process is currently increasing in popularity as a novel solidification casting technique [11-16]” may be unclear due to the reviewer's limited knowledge. During the diffusion-based sintering process, at least one solid fest phase must be present. However, the casting process necessitates complete melting followed by solidification. Perhaps, the authors have redefined the meaning of SPS based on the groundbreaking results derived in this article.

3. Line 51: As emphasized by the authors, the reduction of “core losses” is a key aspect to enhance the efficiency of the soft magnetic materials. Nevertheless, in Section 3 (Results), the power losses of different samples are barely demonstrated.

4. Line 64: Mn and Ni segregation can often be observed in alloys, affecting different properties. Did the authors also investigate the distribution of different elements in the samples produced by means of various fabrication methods in this article?

5. It is recommended to examine the oxygen levels in different samples. When using different manufacturing techniques under various processing conditions, the oxygen content of the resultant samples could significantly differ from each other, which could be a key factor causing different properties.

6. Line 100: It is highly suggested to list the manufacturers and the initial particle sizes of the raw materials.

7. Line 114: Could the authors elaborate on the measurements on the “superconducting quantum device”? What kind of sample geometry is required for the measurements? 

8. Line 127: Regarding the Archimedes method, did the authors place the samples in water or organic solutions? Did the authors observe oxidation (rust) after the oxidation? Why did the samples prepared using different manufacturing techniques exhibit various theoretical densities?

9. Page 4: How did the authors determine the particle size and sphericity after milling?

10. Figure 3 is not mentioned in the text. Could the authors elaborate on the results shown in Figure 3?

11. What is the meaning of FCC and BCC in Figure 3 and Figure 4? It should be clearly clarified that “FCC and BCC” refer to the solid solution (instead of BCC-Fe, FCC-Ni, or BCC-Mn). To the knowledge of the reviewer, the broadening and shifting of an XRD peak can be caused by changes in grain size and internal stresses. Since XRD is a semi-quantitative analysis method, the reliability of the phase fraction shown in Figure 4 b is highly questionable. Could the authors use other more quantitative analytical methods (e.g., EBSD of the powder cross section) to support the results shown in Figure 4b?

12. Could the authors elaborate on the determination of the “crystallite size“ shown in Figure 5 and the lattice constant in Figure 7? Owing to the nature of XRD, could authors try to validate the results in Figure 5 using more accurate analytical methods?

13. Figure 5: If the reviewer understands correctly, the first formation of solid solution suddenly appears after 25 h of milling. What are the crystal phases between 10 h and 25 h of milling? It seems that the majority of the quantitative results is based on XRD. Further studies are highly needed to determine the different phases. Just showing XRD results is not enough.

14. Line 271: Did the authors also investigate fatigue behaviors? Are the “fatigue characteristics“ important for soft magnetic materials?

15. Line 265: Why is the FCC solid solution the single phase in the arc melted samples?

16. The results shown in Figure 11 and Table 1 are very interesting. The reviewer did not really understand, why the “are melting” samples and the “mechanical alloying” showed such high difference in magnetic properties. Perhaps, it could be more understandable after a full writing revision.

17. Could be authors show some images (e.g., SEM, EDS-mapping, EBSD to show grains, element distribution and pores) of the cross-section (instead of the fracture surfaces) of the samples?

Comments on the Quality of English Language

1. The manuscript contains numerous writing mistakes and requires extensive revisions to improve its academic writing quality.

Author Response

We agree the comments

  1. The manuscript contains numerous writing mistakes and requires extensive revisions to improve its academic writing quality.

Answer: The whole article was checked by native English speaker. We hope it fulfills now the desired requirements.

  1. Line 46: The sentence “The spark plasma sintering process is currently increasing in popularity as a novel solidification casting technique [11-16]” may be unclear due to the reviewer's limited knowledge. During the diffusion-based sintering process, at least one solid fest phase must be present. However, the casting process necessitates complete melting followed by solidification. Perhaps, the authors have redefined the meaning of SPS based on the groundbreaking results derived in this article.

Answer :

The reviewer is right, the expression solidification casting technique is unfortunate. The right term is, of course, “consolidation technique”. Thus, we changed this sentence for:

The spark plasma sintering process is currently increasing in popularity as a novel powder consolidation technique [11-16].

  1. Line 51: As emphasized by the authors, the reduction of “core losses” is a key aspect to enhance the efficiency of the soft magnetic materials. Nevertheless, in Section 3 (Results), the power losses of different samples are barely demonstrated.

Answer: We agree the referee comment. We clarify that with mechanical alloying we produce semi-hard material. These materials have also magnetic applications. W add a sentence in the introduction and clarify it in the results section.

Hardening induced by increased microstrain favors the development of a semi-hard behavior. These materials are suitable for magnetically coupled devices, such as the brakes, clutches, and tensioners.

soft (Hc < 1000A/m (12.56 Oe)), semi-hard (10000 A/m >Hc > 1000A/m), hard (Hc > 10000A/m) and superparamagnetic (Hc~0 A/m) behaviors [60]. 

24.7 Oe, corresponding to a semi-hard behavior.

  1. Line 64: Mn and Ni segregation can often be observed in alloys, affecting different properties. Did the authors also investigate the distribution of different elements in the samples produced by means of various fabrication methods in this article?

Answer: This aspect was not addressed in this article and may be part of future works.

  1. It is recommended to examine the oxygen levels in different samples. When using different manufacturing techniques under various processing conditions, the oxygen content of the resultant samples could significantly differ from each other, which could be a key factor causing different properties.

Answer: We indicated in the materials and methods section that the SEM was equipped with a VegaTescan energy dispersive X-ray spectrometry (EDS) analyzer. We didn't notice the oxygen contamination during milling because we were using an argon environment during the mechanical alloying procedure. Neither is the case with the sample obtained by Arc melting under argon atmosphere, as shown in Figure 9. Finally, we suppose that the oxidation in SPS samples is caused by the oxidation of the particle powder after milling when exposed to air during the SPS process, or that the oxidation occurs after the mechanical polish.

  1. Line 100: It is highly suggested to list the manufacturers and the initial particle sizes of the raw materials.

Answer: The modifications were made.

  1. Line 114: Could the authors elaborate on the measurements on the “superconducting quantum device”? What kind of sample geometry is required for the measurements? .

Answer: The measurements ere elaborated on samples with rectangular parallelepiped shape.

  1. Line 127: Regarding the Archimedes method, did the authors place the samples in water or organic solutions? Did the authors observe oxidation (rust) after the oxidation? Why did the samples prepared using different manufacturing techniques exhibit various theoretical densities?

Answer: Water was used for Archimede’s technique. The measures are short (few minutes) and the samples dried after measurement. No oxidation was observed. There is an error in the text, thank you for pointing it out. We should have write “The deduced density values…” instead of “Theoretical density values…”. The modification is made. We introduce the density values in the results section and delete in the materials and methods.

  1. Page 4: How did the authors determine the particle size and sphericity after milling?

Answer: The sphericity is used here as a visual term. No particular measures of the sphericity were made, since we considered that this parameter is not important for the properties of the alloy. The particle size was determined from the high-resolution SEM images using the SEM device software.

  1. Figure 3 is not mentioned in the text. Could the authors elaborate on the results shown in Figure 3?

Answer: The reviewer is right, thank you for pointing out this inconsistency. The following paragraph was inserted in the article:

The XRD patterns were analyzed by Rietveld technique, using MAUD software. An example of Rietveld refinement is given in Fig. 3. for four experimental XRD patterns: unmilled powders, milled for 25, 50 and 130 hours.  The goodness of fit (GOF) values of the refinement mentioned on Fig. 3 show that the refinement can be trusted. The Rietveld analysis allowed us to obtain the phases pro-portion, lattice parameters, size of crystallites, microstrain, and their error bars.

  1. What is the meaning of FCC and BCC in Figure 3 and Figure 4? It should be clearly clarified that “FCC and BCC” refer to the solid solution (instead of BCC-Fe, FCC-Ni, or BCC-Mn). To the knowledge of the reviewer, the broadening and shifting of an XRD peak can be caused by changes in grain size and internal stresses. Since XRD is a semi-quantitative analysis method, the reliability of the phase fraction shown in Figure 4 b is highly questionable. Could the authors use other more quantitative analytical methods (e.g., EBSD of the powder cross section) to support the results shown in Figure 4b?

Answer: The reviewer is right, one can use the term BCC-Fe etc. We preferred to use the more general BCC and FCC terms because the Rietveld analysis (depicted in Fig. 3) does not allow to stipulate what kind of solid solution it is but can reveal a BCC or FCC phase. Concerning the result shown in Fig. 4b, this result is based the Rietveld refinement, which is a quite precise quantitative technique in case of powder analysis. Concerning the EBSD of powders, we are convinced the results obtained by this method is not more precise than the Rietveld refinement.

  1. Could the authors elaborate on the determination of the “crystallite size“ shown in Figure 5 and the lattice constant in Figure 7? Owing to the nature of XRD, could authors try to validate the results in Figure 5 using more accurate analytical methods?

Answer: The results presented in Fig. 5 and 6 are driven from the Rietveld refinement of the XRD patterns. We think that our presentation was not clear enough, that’s why we propose to add the reference to Rietveld analysis when announce the Figures 5 and 6.

  1. Figure 5: If the reviewer understands correctly, the first formation of solid solution suddenly appears after 25 h of milling. What are the crystal phases between 10 h and 25 h of milling? It seems that the majority of the quantitative results is based on XRD. Further studies are highly needed to determine the different phases. Just showing XRD results is not enough.

Answer: The solid solution doesn’t appear after 25h of milling since it is based on the previous existing phases. We plotted the evolution of these phases after 25h since after this time the FCC and BCC phases are the only major phases. Indeed, the most of the quantitative results presented here is based on the Rietveld refinement of the XRD patterns, which is a powerful, trustful and widely used quantitative method.

  1. Line 271: Did the authors also investigate fatigue behaviors? Are the “fatigue characteristics“ important for soft magnetic materials?

Answer: We did not address the fatigue behavior in this study. The fatigue characteristics are important for parts subjected to cyclic mechanical loads (in the order of millions of cycles). Up to now we produced small samples, for fatigue study we need hundreds of fatigue specimens. This will be a completely different study.

  1. Line 265: Why is the FCC solid solution the single phase in the arc melted samples?

Answer: We are not sure that we understand the question. The nature of the chemical elements, in the proportions used here, give in equilibrium conditions a single FCC phase. This is a fact. Thermodynamic calculations using Thermocalc software could show this as well as the probable positions of the atoms in the FCC lattice but this particular study is not the purpose of this article.

  1. The results shown in Figure 11 and Table 1 are very interesting. The reviewer did not really understand, why the “are melting” samples and the “mechanical alloying” showed such high difference in magnetic properties. Perhaps, it could be more understandable after a full writing revision.

Answer: The synthesis of the arc-melted and MA alloys are completely different, with different phase composition. Moreover, the grain size, the grain boundary proportion, the size of crystallites, the microstrain, the density of defects are completely different. This is the reason why these alloys are so different. All these elements are evoked in the sentence “However, the change in magnetic properties can be related to the microstructural changes in the sample after each metallurgical process, as measured by the XRD technique.”

  1. Could be authors show some images (e.g., SEM, EDS-mapping, EBSD to show grains, element distribution and pores) of the cross-section (instead of the fracture surfaces) of the samples?

Answer: The SEM images of the samples before compression tests are shown Fig. 10. One can clearly see the pores. We did not proceed to EBSD analysis. The EBSD investigation would be of course very interesting but overpass the aim of this article.

Comments on the Quality of English Language

The manuscript contains numerous writing mistakes and requires extensive revisions to improve its academic writing quality.

Answer: We check English.

Reviewer 3 Report

Comments and Suggestions for Authors

How was the particle size segregated for spark laser sintering (SLS)? Was there a particle size limitation from maximum and from minimum?
No detailed explanation of how the mechanical alloying (MA) + SPS process was conducted. A process diagram or technical description is missing.
The double designation of the processes as (MA) introduces misunderstanding into the description of the research. (Line 95).
What is the cause of the complete vanishing of the BCC phase when the pressure is increased to 75 MPa and the temperature remains at 750°C?

  Comments on the Quality of English Language

Minor editing of English language required
 - Passive voice misuse
- Punctuation in compound/complex sentences

Author Response

We agree the comments

How was the particle size segregated for spark laser sintering (SLS)? Was there a particle size limitation from maximum and from minimum?

Answer: We are afraid we do not understand the question. There is not spark laser sintering in this study. If the reviewer refers to the Spark Plasma Sintering (SPS), there is no particle size limitation for the use of SPS, but the properties of the final consolidated material is probably dependent of the particles size.

No detailed explanation of how the mechanical alloying (MA) + SPS process was conducted. A process diagram or technical description is missing.

Answer: We addressed bibliographic references for MA and SPS. Please refer to these articles for practical details. The purpose of this article is not to describe these well-known processes, but to study their effect on the alloy properties.

The double designation of the processes as (MA) introduces misunderstanding into the description of the research. (Line 95).

Answer:  This is not the same designation: one is AM while the other is MA.

What is the cause of the complete vanishing of the BCC phase when the pressure is increased to 75 MPa and the temperature remains at 750°C?

 Answer: The SPS is led at high temperature and high pressure. The fact that phase changes are depending on the pressure and not only the temperature is a common behavior of materials. The best simple example is the frozen water melting under 0°C at high pressures. Another hypothesis suggests that the total disappearance of the BCC phase when pressure is raised is due to the phase transition from BCC to FCC. Articles that support this hypothesis have been added to the manuscript [55-57].

Comments on the Quality of English Language

Minor editing of English language required
 - Passive voice misuse
- Punctuation in compound/complex sentences

Answer: we check English.

Reviewer 4 Report

Comments and Suggestions for Authors

The study presents the results of the investigation of the structural, mechanical and magnetic properties of the Fe65Ni28Mn7 alloy obtained by two methods, namely mechanical alloying and spark plasma sintering.

Considering two temperatures, respectively two pressures during the alloy formation procedure, the collected data allows monitoring the changes in the properties of the mixture, and correctly interpreting the results, it is possible to identify how the final product is formed. This information is of real use in the analysis of the obtaining method, as presented by the authors of this study.

The work is original, well organized and easy to understand. The conclusions are supported by the experimental data adequately discussed in the paper.

Minor issues identified:

1. I do not agree with the statement "the disappearance of XRD pattern peaks can be attributed..., lattice distortion". The distortion of the network would have the effect of moving the peaks, not their disappearance.

2. The increase in the network parameter at the time of substitution of Fe and Ni with Mn is due to the change in the electronic configuration. However, a quick explanation can be related to the modification of the atomic radius, as presented by the authors.

3. At line 244 "b is the Burgers vector" shoul be " b is the magnitude/modulus of Burgers vector"

4. At line 316 "100°C under 50 MPa." should be "1000°C under 50 MPa.". Also, the XRD pattern and SEM for sample SPS-1000C 75MPa is missing from figure 10. 

I believe that the study can be considered for publication, after fixes minor issues.

Author Response

We agree the comments

Minor issues identified:

  1. I do not agree with the statement "the disappearance of XRD pattern peaks can be attributed..., lattice distortion". The distortion of the network would have the effect of moving the peaks, not their disappearance.

Answer: statement corrected.

  1. The increase in the network parameter at the time of substitution of Fe and Ni with Mn is due to the change in the electronic configuration. However, a quick explanation can be related to the modification of the atomic radius, as presented by the authors.
  2. At line 244 "b is the Burgers vector" shoul be " b is the magnitude/modulus of Burgers vector"

Answer: statement corrected.

  1. At line 316 "100°C under 50 MPa." should be "1000°C under 50 MPa.". Also, the XRD pattern and SEM for sample SPS-1000C 75MPa is missing from figure 10. 

Answer: There is no sample SPS-1000C 75MPa , there is three samples SPS-750C 50MPa, SPS-750C 75MPa and SPS-1000C 50MPa

Round 2

Reviewer 1 Report

Comments and Suggestions for Authors

The corrections pointed out in the previous review have been made to a large extent. However, the following contradictory statement should be corrected.

In line 269 In the expression "close-packed plane is [110]", it is the direction, not the plane. Please check your expression.

Author Response

We agree the comment of the reviewer.

We modify the manuscript to change plane by direction.